# Irradiation of W and K-Doped W Laminates without or with Cu, V, Ti Interlayers under a Pulsed 6 MeV Electron Beam

**DOI:** 10.3390/ma15030956

**Published:** 2022-01-26

**Authors:** D. Ticoș, M. Galaţanu, A. Galaţanu, M. Dumitru, M. L. Mitu, N. Udrea, A. Scurtu, C. M. Ticoș

**Affiliations:** 1National Institute for Laser, Plasma and Radiation Physics, Atomistilor Street 409, Magurele, 077125 Ilfov, Romania; dorina.toader@inflpr.ro (D.T.); marius.dumitru@inflpr.ro (M.D.); nicoleta.udrea@inflpr.ro (N.U.); adrian.scurtu@inflpr.ro (A.S.); catalin.ticos@inflpr.ro (C.M.T.); 2National Institute of Materials Physics, Atomistilor Street 405 A, Magurele, 077125 Ilfov, Romania; magdalena.galatanu@infim.ro (M.G.); gala@infim.ro (A.G.)

**Keywords:** electron beam, tungsten, irradiation, surface damages, laminates

## Abstract

Small multilayered laminated samples consisting of stacks of W (or K-doped W) foils without an interlayer or with interlayers from Cu, V, and Ti were exposed to a pulsed electron beam with an energy of 6 MeV in several irradiation sessions. All samples maintained their macroscopic integrity, suggesting that the W-metal laminate concept is compatible with high heat flux applications. The surface of the samples was analyzed using a scanning electron microscope (SEM) before and after each irradiation session. The experimental results indicate that electron beam irradiation induces obvious modifications on the surface of the samples. Morphological changes such as the appearance of nanodroplets, nanostructures, and melting and cracking, depending on the sample type and the electron beam fluence, are observed. The irradiation is carried out in a vacuum at a pressure of 2 to 4 × 10^−2^ torr, without active cooling for the samples. The structures observed on the surface of the samples are likely due to electron beam heating and vaporization followed by vapor condensation in the volume adjacent to the surface.

## 1. Introduction

The nuclear fusion produced in magnetically confined plasmas stands out as the most promising green candidate for future energy generation [1]. The selected ITER plasma-facing materials have been tested at different facilities that recreate some of the conditions attained in fusion machines [1,2]. These facilities use powerful plasma guns or linear plasma devices to produce high-density plasma [3,4,5,6,7,8,9], electron beams [10,11,12,13,14], and ion beams (mostly H+ and He+) [8,12,15].

Taking into account the extreme conditions that the plasma-facing armor materials are subjected to, only a few materials may be suitable: carbon fiber composite (CFC), tungsten, beryllium and lithium. The disadvantage of the CFC is its enhanced erosion due to chemical interaction with hydrogen plasma. Exposures of the carbon fiber composites by multiple plasma pulses were performed at plasma gun facilities. The experiments and simulations have shown that the CFC erosion is about a few microns per shot [16]. The investigation of the erosion as induced by deuterium ions in beryllium, tungsten, and carbon shows that the sputtering yield for tungsten is the lowest, being almost one order of magnitude lower than that of carbon for energy ranges between 100 and 10 keV [17,18]. Thus, tungsten is the main candidate for building the plasma-facing armor components for future fusion reactors due to its high melting temperature (3695 K), low tritium inventory, low sputtering yield for tritium nuclei [19,20], and high threshold energy for sputtering by deuterium (200 eV). The weakness of tungsten is its brittleness below the ductile-to-brittle transition temperature (420–670 K) and the reduction in its strength and hardness above the recrystallization temperature (1300–1600 K) [21].

One of the critical issues in the operation of a tokamak plasma is the production of runaway electrons, associated with the disruptive events [22]. The energy of such high-energy electrons is in the range from MeV to tens of MeV, while the current can reach several MA. Such fluxes of intense and energetic electrons can easily destroy the first wall and damage the machine. The behavior of materials under such particle bombardment is interesting and hardly reproducible in testing facilities. Most testing facilities produce electron beams with energy in the range of 40–200 KeV, at powers ranging from 60 kW up to ~1 MW [15]. Here, we utilize an electron beam (EB) with an energy of 6 MeV obtained from a linear electron accelerator, at a power of up to ~95 kW and delivered in pulses of 4 µs at a frequency of ~50 Hz [23,24].

There is an ongoing, persistent quest to find the most suitable fusion materials and novel solutions employing W, such as W-fiber components, W laminates or “smart” W alloys [25,26,27] or to join such materials in components [28]. The goal is to remove the heat as efficiently as possible from the surface exposed to the heat load of the plasma.

In this report, we analyze so-called W-laminate materials, i.e., multi-layered composites either from alternate W and other metal foils stacked and joined together or even from only W foils without other interlayers. The main idea behind the laminate concept is to transfer as much as possible from the excellent mechanic properties of thin (severely deformed) W foils (with a thickness of around 0.1 mm) to bulk materials [29]. The composites with W and other metals (Cu, Ti, V) are potential candidates for the divertor cooling system, while those made only from W foils (either pure W or K-doped W with a higher recrystallization temperature) might be used both as components in the cooling system or as armor material. Specimens from such W laminates were irradiated with the 6 MeV electron beam in vacuum at 2–4 × 10^−2^ torr and without active cooling, as a screening test for future typical high heat flux tests to be performed in facilities such as JUDITH (FZJ) or GLADIS (IPP). The irradiation of the samples was carried out in one to three sessions, each lasting 10 min. The study focused on laminates’ macroscopic integrity and the surface modifications induced by the EB. The heating, vaporization, and condensation of the vapors led to the formation of small nanometric-sized structures, partially covering the surface of the samples. Their surface morphology was analyzed under a Scanning Electron Microscope (SEM). Each sample surface was analyzed before irradiation and after each irradiation session.

## 2. Materials and Methods

### 2.1. Samples Structures

Several types of multi-layered samples were prepared by Field Assisted Sintering Technology (FAST) [30], as shown schematically in Figure 1. The W-Cu, W-Ti, and W-V laminates are made from pure W foils with a 100 ± 10 µm thickness, Cu and Ti foils with a thickness of 100 ± 10 µm as well, while the V foils have 127 ± 10 µm thickness. Due to the strong inter-diffusion of W and Ti, for the W-Ti laminate, a thin Cr interlayer was deposited by RF sputtering on the W foils, with a thickness of 150 ± 10 nm. Essentially, the laminates consist of alternating foils W/Cu/W/Cu…, W/V/WV… and in the case of the W-Ti laminate, the sequence is W + Cr/Ti/Cr + W + Cr/Ti….

Single-material samples were also used, one made exclusively of pure W thin foils with a 100 ± 10 µm thickness and two samples made of K-doped W foils with thicknesses of 100 ± 10 µm and 220 ± 15 µm, where the W had about 50 to 100 ppm of K in their composition. All samples had a square shape about 10 × 10 mm and a thickness of 4–5 mm (40 foils, each). 

The main advantage of the FAST method (also termed as SPS or Spark Plasma Sintering) is the short processing time, lasting just a few minutes at high temperatures, resulting in lower recrystallization detrimental effects. In the case of bulk metallic samples, the main heating arises in FAST through the Joule effect. When putting together stacks of foils, at the interfaces between these foils, due to imperfect electrical contacts, the electrical discharges produce effects similar to electrical point welding, but on the entire surface immediately. As the interfaces tend to improve due to the heating- and discharge-stimulated mass transport, a diffusion bonding process also starts [30]. The joining was performed at temperatures ranging from 860 °C for W-Cu laminates up to 1350 °C for K-doped W laminates. The dwell time was around 6 min for all samples made with pure W foils, while for those with K-doped W foils, it was extended to about 20 min. Note that the temperatures mentioned above are not measured exactly at the sample level, but at the top of the graphite piston. 

High-resolution imaging of the W surface in the pristine samples was performed by an FEI Inspect S50 SEM. The Everhart–Thornley detector (ETD) was used to work with secondary electrons (SE).

The analysis of the W surface of the pristine samples is shown in Figure 2 and Figure 3. The operating high voltage was 20 kV, while the current was ~20 pA.

The roughness of the surface is due to the fabrication process. All samples were mechanically polished using a diamond grinding machine; however, some types of cracks and extrusions were left, as can be seen in the images of Figure 2 obtained by FEI Inspect S50 SEM analysis. Additionally, scattered micron-size grains can be seen on the surface. They are clearly visible at higher magnification. In the images of Figure 3, the surface of the three single-layer samples of W, (W-K)_1_ and (W-K)_2_, are shown.

### 2.2. Irradiation Procedure

Irradiation of the samples was carried out in three sessions, each lasting 10 min. After each irradiation session, the samples were left to cool down for about 20 min in vacuum to the ambient temperature and were removed from the vacuum chamber for further analysis. Their non-irradiated surface was analyzed under a SEM after each cooling, and then subjected again to another irradiation session.

In the irradiation setup, the W layer is in an upward position, as shown in Figure 1, facing the EB, which is emitted from the LINAC downwards towards the ground. The EB fluence was measured with a Faraday Cup Radiabeam FARC-04-2M. The charge per pulse was directly obtained by integrating the voltage curve for the time produced by the cup [31]. In the irradiation sessions 1 and 3, the spot size of the electron beam showed a characteristic asymmetric Gaussian profile, approximately elliptical with axes of 18 by 14 mm. The fluence per pulse was $6.2\times 10^{10}$ and $1.4\times 10^{11}$ el/cm^2^, respectively.

In the second irradiation session, a set of 2 identical quadrupoles Radiabeam EMQR-01-158-240 were inserted in the beam line, in order to make the EB spot larger and more uniform on the surface of the samples. The fluence per pulse in this second session was $7.3\times 10^{10}$ el/cm^2^. The LINAC delivered EB pulses at a frequency of 53 Hz and pulse duration of 4 µs. A single irradiation session was applied to the K-doped W and pure W laminates with a fluence per shot of $1.6\times 10^{11}$ el/cm^2^. At the highest fluence, the peak EB current density per pulse crossing the samples was 6.4 mA/cm^2^. The timetable of irradiation sessions for each sample is presented in Table 1.

The total energy of the electron beam is $W_{t}=N_{e}\cdot S\cdot E_{e}$, where *N_e_* is the total electron fluence (given in Table 1) for one irradiation session, *S* is the surface of the sample (in cm^2^), and $E_{e}=9.6\times 10^{-13}$ J (corresponding to 6 MeV) is the beam energy. For the first irradiation session, the electron beam fluence is $1.9\times 10^{15}$ el/cm^2^ and the total energy on the sample surface is 1824 J, while at $2.3\times 10^{15}$ el/cm^2^, $4.4\times 10^{15}$ el/cm^2^ and $5\times 10^{15}$ el/cm^2^, the energy deposition is 2208 J, 4224 J, and 4800 J, respectively.

## 3. Results and Discussion

### 3.1. First Irradiation Session of Multi-Layer Laminates at the 6 MeV LINAC

After 10 min of irradiation, $1.9\times 10^{15}$ el/cm^2^ of cumulated EB fluence on the surface of the samples, viewed on a FEI Inspect S50 SEM with a magnification of ×20,000 and an accelerating voltage of 20 kV, is marked by some visible morphological changes. The W-Ti sample surface becomes granular in appearance, with granules measuring, on average, ~1 µm, as shown in Figure 4a. The W-Cu sample surface does not show any distinguishable differences after irradiation (Figure 4b). For the W-V sample, morphological changes can be seen in the image in Figure 4c. It appears that the heating by bombardment with the 6 MeV electron beam led to the formation of small nanometer-size particulates, well below 1 µm, which cover the surface non-uniformly. 

### 3.2. Second Irradiation Session of Multi-Layer Laminates at the 6 MeV LINAC

In the second irradiation session, the total cumulated EB fluence was $2.3\times 10^{15}$ el/cm^2^ over a time period of 10 min, corresponding to an exposure to 31,800 consecutive EB pulses.

The high-resolution imaging of the W surface after the second session was performed by an Apreo S SEM made by ThermoFisher Scientific, with the same type of detector (i.e., EDT) working with secondary electrons. The accelerating voltage used was 10 kV, and the current was 25 pA.

Interesting morphological changes are seen now, as shown in the images of Figure 5. In Figure 5a, the texture surface of the W-Ti sample shows the formation of dendrites, with the average size of the structures being about 500 nm. In Figure 5b, one can see on the W-Cu sample the appearance of tiny nano-particulates that are 10–300 nm in size and have different shapes (spherical and cylindrical). The surface of the W-V sample presented in Figure 5c shows the melting and solidification of the exposed surface due to the EB heating, with nanometer-size droplets spread over the analyzed area.

### 3.3. Third Irradiation Session of Multi-Layer Laminates at the 6 MeV LINAC

The cumulated fluence in the third irradiation session was $4.4\times 10^{15}$ el/cm^2^, corresponding to the same 10 min period.

Several morphological changes can be seen from the previous irradiation session, as shown in Figure 6. One can observe that all sample surfaces are covered with a relatively dense layer of grains. Uniform and dense particle distribution can be seen on the surface of the W-Ti sample, and grain agglomeration together with cracks on the W-V sample surface. Interestingly, in the case of W-Cu, the shape of these structures is mainly cylindrical, appearing as bright elongated chips on the exposed surface, as seen in Figure 6b. It appears that the heating, superficial vaporization, and condensation of the vapors led to the formation of these small nanometric-size structures, covering the surface of the samples.

### 3.4. Single Irradiation Session of the K-Doped W and Pure W Laminates at 6 MeV

We exposed the single K-doped W and pure W laminates to a cumulated flux of $5\times 10^{15}$ el/cm^2^, corresponding to a single 10-min irradiation period. The results from the Apreo S SEM are shown in Figure 7.

One can clearly notice that the whole surface of the bare W sample session is covered by spherical droplets with a wide range of sizes, between 10 and 200 nm, as shown in Figure 7a. The K-doped laminate surfaces are also covered by agglomerations of larger nanometer-size particles with different shapes (mostly spherical). Their size varies in range, from tens of nanometers in the case of (W-K)_1_ to much smaller nanometer particles in the case of (W-K)_2_. One can also see intragranular micro-cracks, primarily perpendicular to the sample surface in the case of (W-K)_1_. 

An energy-dispersive spectroscopy (EDS) analysis reveals the composition of some droplets, as shown in Figure 8. The EDS detector used was an Octane Elite Super of the Element system from EDAX.

The EDS spectra of the W-K laminate droplets are shown in Figure 9. The peaks shown are only the X-ray emission lines of W, i.e., M_α1_ = 1.776, M_α2_ = 1.774, M_β_ = 1.83, L_I_ = 7.38, L_α_ = 8.4, L_β1_ = 9.67, L_β2_ = 9.96, L_γ1_ = 11.29, and L_γ3_ = 11.67. The EDS analysis showed that in all three cases, the nanosized droplets seen on the surface are composed only of W. The agglomeration of the particles is due to the thermal effects on the irradiated sample. We believe that these droplets are induced by strong surface heating followed by the superficial melting and cooling of the material vapors.

### 3.5. Discussion

It should be noted that for W, the total stopping power of an electron with an energy of 6 MeV, is S = 1.798 MeV/g cm^2^, as tabulated in reference [32]. Given the W density, $\rho_{W}=19.3$ g/cm^3^, one can roughly approximate the loss of energy for an electron crossing the pure W laminate as $S\cdot \rho_{W}=3.4\;\mathrm{MeV}/\mathrm{mm}$. Moreover, the penetration range of electrons (in the continuous slowing down approximation) is $R_{CSDA}=2.21$ mm, as provided by the table of reference [32]. The EB deposits most of its energy in the whole volume of the samples; therefore, the heating also strongly affects the non-irradiated surface. The other materials, while having almost similar stopping power values, differ from W by their density. Cu and Ti have the same total stopping power, ~1.69 MeV/g cm^2^, while K and V have stopping powers of ~1.84 MeV/g cm^2^ and ~1.66 MeV/g cm^2^, respectively. Their density is, however, smaller than that of W by a factor of 2 to 4: $\rho_{Ti,V,Cu}=4.5,\;6.1,\;8.96$ g/cm^3^, respectively; therefore, the dominant behavior of heat absorption from the EB is established by W. In the case of K (which is at a concentration of 50–80 ppm and dispersed in W), the density of $\rho_{K}=0.862$ g/cm^3^ is smaller by a factor of 22 than that of W, and its overall contribution to the stopping power of electrons is negligible. An insignificant contribution to the stopping power of the EB is also made by Cr, due to its greatly reduced thickness. The melting and boiling temperatures of K are much lower than those of W, and this explains why no K droplets were observed in the K-doped W laminates, as any K material on the surface is vaporized beforehand. Additionally, the melting point of Cu, Ti and V is lower than that of W, but no droplet formation was observed outside of the materials (on the laterals of the samples). 

We can make an estimate of the heating temperature of the laminates by using $\;\Delta T_{heat}\approx W_{t}/\underset{i}{\sum }m_{sample\;\left(i\right)}⬝c_{i}$, where $m_{sample\;\left(i\right)}$ is the mass of the laminate type (i) and $c_{i}$ is its specific heat. This evaluation excludes any thermal losses by direct contact of the samples with the support (which was made of stainless steel).

As an example, in the case of the W-Cu laminate, we obtain $\Delta T_{heat}=1531$ °C at the lowest fluence ($1.9\times 10^{15}$ electrons/cm^2^), considering the specific heats of W ($c_{i}$ = 0.13 J/g K) and Cu ($c_{i}$ = 0.385 J/g K). For the highest fluence, i.e., for $4.4\times 10^{15}$ electrons/cm^2^, $\;\Delta T_{heat}=3546$ °C. However, we do not observe the vaporization of the Cu part of the laminate or loss of its integrity; therefore, the temperature of the full sample is well below this estimated value. Moreover, at the lowest fluence, no modifications are observed on the surface of the samples, as shown in Figure 4. This situation changes after the second and third irradiation sessions, during which the EB fluence (and incident energy) is increased by ~21% and ~131%. In these cases, morphological changes are clearly seen on the surface, consisting of local vaporization and condensation, as shown in Figure 5 and Figure 6. 

From the point of view of irradiation, the difference between the samples is determined by how much heat is contained in the irradiated sheet. While in W-W and W-Cu the heat is dissipated more easily due to the high thermal conductivity of W and Cu (182 W/mK and 401 W/mK, respectively), in the case of samples with V and Ti, the conductivity of the layers is low (31.3 W/mK and 22.4 W/mK, respectively). This is particularly visible after the first irradiation session at the lowest fluence, when the W-Cu surface appears unaffected, while those of W-V and W-Ti show signs of melting or nanoparticle formation. However, at higher fluences, the surface damages shown in Figure 5b and Figure 6b in the W-Cu laminate, as well as in the W-V and W-Ti laminates, in spite of their different morphologies, indicate that a very intense heating takes place in the near vicinity of the irradiated surface, before the heat is dissipated across the sample. Moreover, as shown in [30], temperatures exceeding the melting point of the metal result in a dramatic deterioration of the entire material. On the other hand, the inter-diffusion of W and the other metals can be excluded due to the much larger time-scale of diffusion processes. Based on these observations, we assert that the dendrites, droplets, and elongated chips seen in Figure 5 and Figure 6 are made entirely of W with no phase mixing with V, Cu and Ti in the vaporization process. Mixed composite nanoparticles made of W and Cu or W and Ti were observed to form in rather extreme experimental conditions when compared to ours, consisting of a very rapid heating produced by a pulsed current with a peak value in the tens of kA passing through the W/Cu/Ti samples, which led to the explosion of the samples, followed by compression at 1.5 GPa, as reported in [33,34]. Additionally, these experiments took place at atmospheric pressure after adding Ar as a buffer gas, as opposed to vacuum as reported here. 

Nanoparticle formation has been reported in other works, where W was irradiated with a pulsed EB with energy 120 keV [35] and He+ ions with energy in the few keV and a fluence similar to our experiment [36]. Interestingly, nano-dendrites and even nano-tendrils were observed to form on the surface of W irradiated with He+ ions having an energy above a few hundred eV and up to 12 keV [37,38], while we observed some dendritic structures after irradiation with electrons at 6 MeV. In light of our findings, all samples passed the irradiation tests successfully; however, the superior thermal conductivity of the W-Cu and pure W laminates should be taken into account. Additionally, the W-Cu laminate did not show any cracks on its surface after the irradiation sessions and appears to be more robust.

## 4. Conclusions

The irradiation of laminates made of alternating layers of W, Cu, V, Ti and K-doped W layers was carried out in vacuum with the electron accelerator ALID 7 at total beam fluencies between 1.9 × 10^15^ and 5 × 10^15^ el/cm^2^ in sessions of 10 min. The laminates W-Ti, W-Cu and W-V were irradiated in each of the three sessions. The K-doped W laminate and a laminated sample made of pure W layers were exposed to a single 10-min irradiation session. All samples maintained their macroscopic integrity. SEM analysis of the sample surfaces revealed important morphological changes due to surface and volume heating by the EB, especially at higher beam fluences. The EDS analysis showed that the droplets were made entirely of W on the K-dopped W laminates surfaces. Uniform and denser nanometric particle distribution on the surface, particle agglomeration, different sizes and shapes of structures covering the surface non-uniformly and cracks are observed upon the EB heating and vaporization of the surface, followed by vapor condensation in the area adjacent to the surface. All laminates withstood the irradiation tests; however, the W-Cu laminate did not show any surface cracks. Since the irradiation was performed without active cooling of the samples, the heat accumulates in the material at a much higher rate than in the expected operating conditions (with active cooling).

## Figures and Tables

**Figure 1 materials-15-00956-f001:**
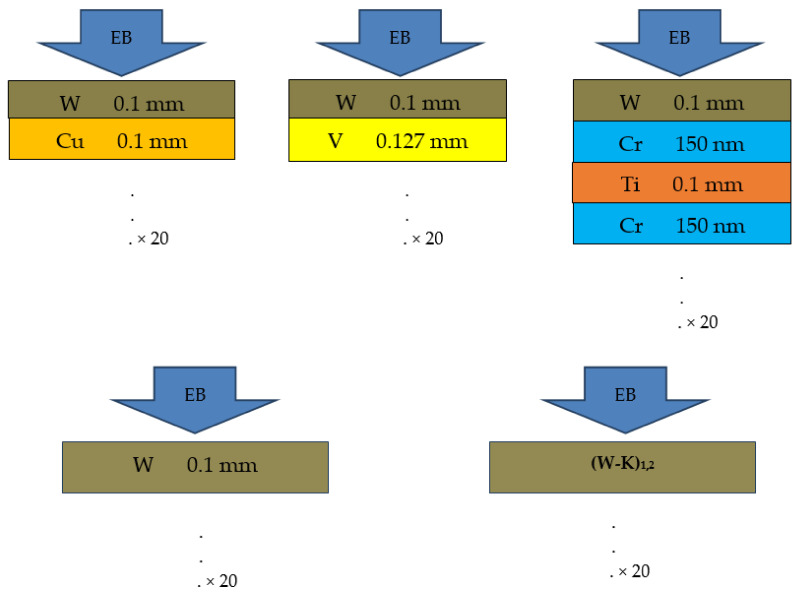
Sketch of the structure of the W-Cu, W-V, W-Ti, and pure W and K-doped W laminates (not to scale). The laminates consist of alternating foils W/Cu/W/Cu…, W/V/WV…, W + Cr/Ti/Cr + W + Cr/Ti… with a total of 40 foils. The K-doped W laminates were produced from 100 and 220 µm foils, for (W-K)_1,2_, respectively. The samples are irradiated with the EB perpendicular on the top W layer.

**Figure 2 materials-15-00956-f002:**
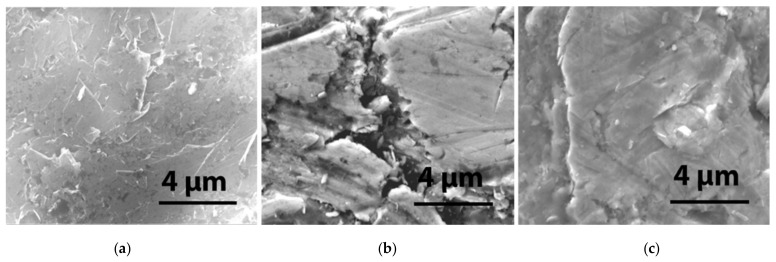
High-magnification images (×20,000) of the W surface in the pristine samples: (**a**) W-Cr-Ti; (**b**) W-Cu and (**c**) W-V.

**Figure 3 materials-15-00956-f003:**
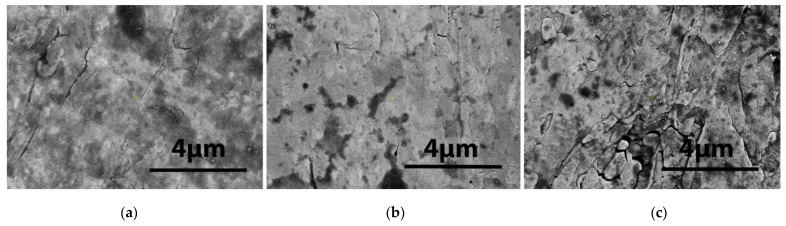
High-resolution images (at the similar magnification as in Figure 2 of the samples made of (**a**) W and (**b**) (W-K)_1_ laminate and (**c**) (W-K)_2_ laminate.

**Figure 4 materials-15-00956-f004:**
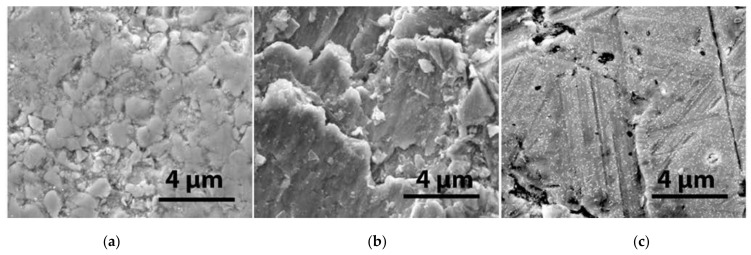
High-magnification images (×20,000) of W surface after first session of 10 min irradiation in the 6 MeV EB: (**a**) W-Ti; (**b**) W-Cu and (**c**) W-V.

**Figure 5 materials-15-00956-f005:**
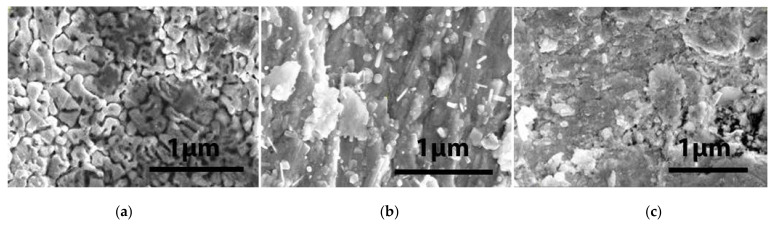
High-magnification images of W surface after the second session of 10 min irradiation at 6 MeV: (**a**) W-Ti (×50,000); (**b**) W-Cu (×50,000) and (**c**) W-V (×50,000).

**Figure 6 materials-15-00956-f006:**
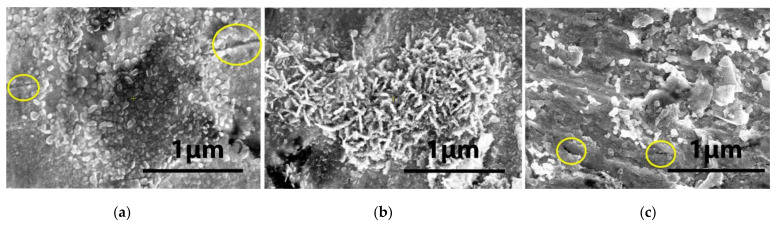
High-magnification SEM images of W surface after the third session of 10 min irradiation at 6 MeV: (**a**) W-Ti (×50,000); (**b**) W-Cu (×50,000); and (**c**) W-V (×50,000). Some observed cracks are shown with circles.

**Figure 7 materials-15-00956-f007:**
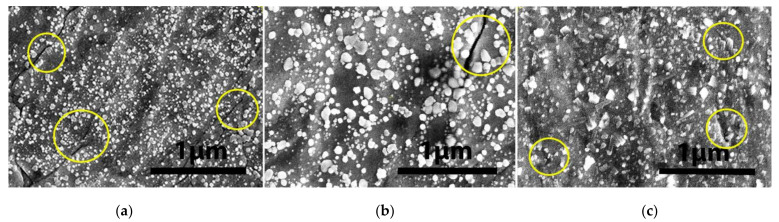
High-magnification SEM images of irradiated samples for a duration of 10 min in the 6 MeV EB: (**a**) pure W laminate (×50,000), (**b**) (W-K)_1_ laminate(×50,000) and (**c**) (W-K)_2_ laminate(×50,000). Large cracks are observed on the surfaces marked with circles.

**Figure 8 materials-15-00956-f008:**
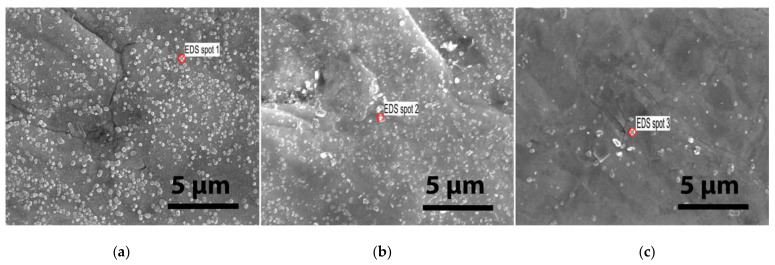
High-resolution SEM images of irradiated samples for a duration of 10 min in the 6 MeV EB and the droplets chosen for an EDS analysis: (**a**) pure W laminate, (**b**) (W-K)_1_ laminate and (**c**) (W-K)_2_ laminate.

**Figure 9 materials-15-00956-f009:**
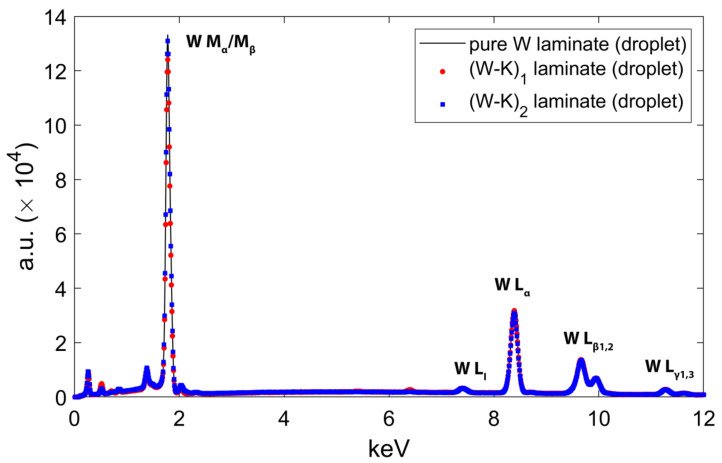
The X-ray spectra of pure W laminate droplet, (W-K)_1_ laminate droplet and (W-K)_2_ laminate droplet.

**Table 1 materials-15-00956-t001:** Table with sample type, irradiation session and total EB fluence during a session. Each session lasts for 10 min.

Sample	Irradiation Session	Electron Beam Total Fluence (el/cm^2^)
W-Cu	1	$1.9\times 10^{15}$
W-Cu	2	$2.3\times 10^{15}$
W-Cu	3	$4.4\times 10^{15}$
W-V	1	$1.9\times 10^{15}$
W-V	2	$2.3\times 10^{15}$
W-V	3	$4.4\times 10^{15}$
W-Ti	1	$1.9\times 10^{15}$
W-Ti	2	$2.3\times 10^{15}$
W-Ti	3	$4.4\times 10^{15}$
W	1	$5\times 10^{15}$
(W-K)_1_	1	$5\times 10^{15}$
(W-K)_2_	1	$5\times 10^{15}$

## Data Availability

The data presented in this study are available on request from the corresponding author.

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
