# Peer review of "Irradiation of W and K-Doped W Laminates without or with Cu, V, Ti Interlayers under a Pulsed 6 MeV Electron Beam"

_materials, 2022, doi:10.3390/ma15030956_

Round 1

Reviewer 1 Report

The authors attempt to analyze the effects of high energy (6 MeV) electron beam irradiation of various exposures on W and K-doped W with and without the the Cu, V, and  Ti interlayers.  The analysis was carried out using SEM imaging to investigate the surface morphology nano-modifications on the samples at various levels of beam fluence. The work is promising in giving an empirical understanding of the material behavior of plasma facing materials such as W laminates used in nuclear fusion machines.  The paper is well written in terms of language usage and structure. The experimental procedure is clearly explained and the overall structure of the paper is good. This reviewer recommends that the paper be recommended for publication after they comply with the following comments: 

  1. In Fig. 1, the authors should include the thicknesses of the W, K-W laminates and their interlayers in the diagrams.  This will facilitate ease in perusal and puts the Figure in proper context of the materials and methods section.
  2. There should be consistency of the magnification scale in all the SEM images.  If there is a zoomed in image, it has to be an inset to that particular image.  For example, In Fig. 5, all three should images should be in 4 micron range and if fig. 5(b) should be zoomed in to 1 micron range then that has to be a separate image put as an inset to Fig. 5(b).  
  3. The analysis employed in this paper rests mainly in the surface morphology modifications observed for various irradiation levels through SEM of the surface.  This method alone cannot fully characterize the mechanism of the nanostructure formation due to the EB irradiation.  The analysis of the samples would benefit largely if it could be coupled with information from the layered structures using X-ray Photoelectron Spectroscopy (XPS) to see how the various irradiation fluence affected the bulk material. Also compositional analysis through Inductively Coupled Plasma Optical Emission Spectroscopy (ICP-OES) could help identify more deeply the origins and mechanisms involved in the formation of these nano-modifications.
  4. The authors should also mention in the discussion section how the structural integrity of each of the various sample configurations are affected by the appearance of the morphological nano-modifications of each sample.  This again bring up the point that maybe there is a need for a cross sectional SEM analysis at the site of irradiation of the samples to completely characterize them.
  5. The authors did not provide a recommendation as to which of the sample configurations they analyzed performed best under the various irradiation dosages. This should be added to the discussion and conclusion sections.

Author Response

We thank the referee for giving us the opportunity to submit a revised draft of the manuscript “Irradiation of W and K-doped W laminates without or with Cu, V, Ti interlayers under a pulsed 6 MeV electron beam” for publication in the Materials Journal. We appreciate the time and effort that the reviewer has put to provide feedback on our manuscript. We would like to thank very much the reviewer for careful and thorough reading of the manuscript and for the thoughtful comments and constructive recommendations, which helped improve the quality of this manuscript. We thoroughly revised the manuscript in order to accommodate the comments of all reviewers. The changes are written in blue within the manuscript. Below is a point-by-point response to the reviewer’s comment. We hope that our revised manuscript is now suitable for publication in the journal Materials.

Reviewer 2 Report

In this work authors studied the changes in the morphology of multilayered laminated samples consisting of stacks of W (or K-doped W) foils resulted from high-energy electron beam explosion. The formation of metal nanoparticles of different size and shape depending of the sample composition was demonstrated. Therefore, this study contributes to materials science and high-energy physics revealing new ways to obtain nanosized metal materials. The article is well-written. The article can be accepted after minor revision. Comments:

  • Page 2, lines 129-132 Author wrote “It appears that the heating by bombardment with the 6 MeV electron beam has led to the formation of small nanometer-size particulates, well below 1 μm, which cover the surface  non-uniformly. “ Please, provide the chemical composition of nanoparticles. Do they consist of single metall or W-V alloy? The same question is addressed to other obtained nanoparticles (sections 3.2 and 3.3). I suggest to analyze the distribution of all elements in a selected area by EDX elemental mapping.
  • Page 4, line 187. Please provide how total stopping power was found. 

Author Response

We thank the referee for giving us the opportunity to submit a revised draft of the manuscript “Irradiation of W and K-doped W laminates without or with Cu, V, Ti interlayers under a pulsed 6 MeV electron beam” for publication in the Materials Journal. We appreciate the time and effort that the reviewer has put to provide feedback on our manuscript. We would like to thank very much the reviewer for careful and thorough reading of the manuscript and for the thoughtful comments and constructive recommendations, which helped improve the quality of this manuscript. We thoroughly revised the manuscript in order to accommodate the comments of all reviewers. The changes are written in blue within the manuscript. Below is a point-by-point response to the reviewer’s comment. We hope that our revised  manuscript is now suitable for publication in the journal Materials. Please see the attachment.

Reviewer 3 Report

The manuscript sheds light on morphology of multilayered laminated samples made from W (K-doped W) and W-Cu, W-Ti, W-V foils and subjected to high-energy (6 MeV) electron beam treatment. The authors provides novel results describing the modification of sample surfaces due to heating, melting and evaporation. However, the manuscript has obvious drawbacks and couldn’t be published in the current form due to the following:

1) The introduction section is well prepared and problem statement is outlined. Nevertheless, it’s not clear why the authors utilized nanolaminate materials instead of bulk materials or thin (nanometer) films. Is it associated with the heat transfer between the layers of the foils? Please, make the scientific arguments of using the multi-layered W foils in the case of electron-beam irradiation. In this regard, the improved motivation of the study of these specific alloys should be added in the abstract.

2) In the «Materials and Methods» section the detailed procedure of sample preparation is required. The cited paper [30] is not enough to demonstrate the technology and layered structure of as-cast materials (before irradiation).  It is advisable to replace Fig. 2 and Fig. 3 into «Results and Discussion» section. In addition, it’s necessary to provide a complete description of SEM analysis technique, namely, type of microscope (microscope brand, manufacturer), accelerating voltage, type of used detectors, beam current, etc.

3) The polishing of samples does induce defects onto the sample surface. However, no information on surface preparation is given in the methodology section. Please, describe the polishing procedure in order to understand whether the surface defects are caused by the fabrication process or mechanical polishing.

4) All shown Figures of SEM analysis do not contain any information about type of detectors (secondary or back scattered electrons). Therefore, it is impossible to distinguish SEM contrast originated from the particles, droplets, grains or other morphology features.

5) Page 2, line 127: «The W-Cu sample surface does not show any distinguishable differences after irradiation…». How do the authors explain the absence of any noticeable changes caused by high-energy electron irradiation of W-Cu?

6) Page 3, line 143: «… the texture surface of the W-Ti sample shows the formation of nano-tendrils with the average size of the structures of about 500 nm». Most likely, the authors reveal the formation of dendritic structure instead of tendrils. To make a conclusion on the morphology of crystals, the analysis of W-Ti phase diagram is desirable, because the electron irradiation leads to surface melting and intermixing of layers, accompanied by formation of fast quenched crystals of different morphology (needles, dendrites, etc.). I am not sure, that interpretation of SEM images given by the authors on the nano-tendril structure is validated.

7) Considering Fig. 6, do the authors see craters on the sample surface? Some cracks are not clearly seen in the images, so the additional graphics or more indicative images are required to show the defects appeared after irradiation.

8) Page 4, line 179: «An EDX analysis has shown that  in all 3 cases the nanosized droplets seen on the surface are composed only of W». If you have an EDX spectrometer installed on the SEM it’s advisable to add the corresponding EDX spectra of droplets and grains to prove the presence of only W atoms into the observed regions.

9) If the electron beam irradiation induce surface melting, did the authors observe any binary phases of W-Cu, W-Ti and W-V phase systems? Phase composition is a very essential issue that could not be completely ignored. Usually, this kind of studies include XRD patterns or EDX analysis of every inclusions formed onto the sample surfaces.

10) Discussion section does not contain any important results of SEM study. The science is born inside the discussion. Without explanation of the observed structure shown in the SEM images this manuscript has very poor scientific meaning for the readership of Materials Journal.

Author Response

(The authors gave the same response as above.)

Round 2

Reviewer 1 Report

The authors put good effort to further improve their manuscript and they have satisfied all the requirements and satisfactorily addressed all the reviewer’s comments. Therefore this reviewer recommends that this manuscript be published in Materials Journal.

Author Response

Thank you for your recommendation that our manuscript be published in Materials Journal.

Reviewer 3 Report

The authors have made a good work to prepare the replies on my comments and the revised manuscript has been improved significanly. So, I recommend this paper for publication. 

Author Response

(The authors gave the same response as above.)
